# Sarcoidosis-Associated Sensory Ganglionopathy and Harlequin Syndrome: A Case Report

**DOI:** 10.3390/medicina59081495

**Published:** 2023-08-20

**Authors:** Ieva Navickaitė, Miglė Ališauskienė, Sandra Petrauskienė, Gintarė Žemgulytė

**Affiliations:** 1Department of Neurology, Medical Academy, Lithuanian University of Health Sciences, A. Mickeviciaus Str. 9, LT-44307 Kaunas, Lithuania; migle.alisauskiene@lsmu.lt (M.A.); gintare.zemgulyte@lsmu.lt (G.Ž.); 2Department of Preventive and Paediatric Dentistry, Lithuanian University of Health Sciences, Luksos-Daumanto Str. 6, LT-50106 Kaunas, Lithuania; sandra.zemgulyte@lsmu.lt

**Keywords:** ganglionopathy, small-fiber neuropathy, Harlequin syndrome, sarcoidosis

## Abstract

*Background and Objectives*: Sensory ganglionopathy is a rare neurological disorder caused by degeneration of the neurons composing the dorsal root ganglia. It manifests as various sensory disturbances in the trunk, proximal limbs, face, or mouth in a patchy and asymmetrical pattern. Harlequin syndrome is characterized by unilateral flushing and sweating of the face, neck, and upper chest, concurrent with contralateral anhidrosis. Here, we present and discuss a clinical case of sarcoidosis-associated ganglionopathy and Harlequin syndrome. *Case presentation*: A 31-year-old woman complained of burning pain in the right side of the upper chest and the feet. She also experienced episodes of intense flushing and sweating on the right side of her face, neck, and upper chest. Three years before these symptoms began, the patient was diagnosed with pulmonary sarcoidosis. On neurological examination, sensory disturbances were present. In the trunk, the patient reported pronounced hyperalgesia and allodynia in the upper part of the right chest and some patches on the right side of the upper back. In the extremities, hypoalgesia in the tips of the fingers and hyperalgesia in the feet were noted. An extensive diagnostic workup was performed to eliminate other possible causes of these disorders. A broad range of possible metabolic, immunological, and structural causes were ruled out. Thus, the final clinical diagnosis of sarcoidosis-induced sensory ganglionopathy, small-fiber neuropathy, and Harlequin syndrome was made. Initially, the patient was treated with pregabalin and amitriptyline, but the effect was inadequate for the ganglionopathy-induced pain. Therefore, therapeutic plasma exchange as an immune-modulating treatment was selected, leading to partial pain relief. *Conclusions*: This case report demonstrates the possible autoimmune origin of both sensory ganglionopathy and Harlequin syndrome. It suggests that an autoimmune etiology for these disorders should be considered and the diagnostic workup should include screening for the most common autoimmune conditions.

## 1. Introduction

Sensory ganglionopathies, also called neuronopathies, are a spectrum of rare acquired or genetic disorders caused by degeneration of the dorsal root ganglia (DRG) and its central and peripheral projections [1]. Nearly a third of sensory ganglionopathy cases are paraneoplastic in origin, while autoimmune etiology represents approximately 20% of cases [2]. A few new ganglionopathy-related antibodies were recently discovered, including antibodies against fibroblast growth factor receptor 3 and argonaute, suggesting that an even higher percentage may be attributed to an autoimmune mechanism [1]. Pathogenetically, the high susceptibility to various blood-derived antibodies, toxins, and infectious agents can be explained by the unique vascularization of the DRG. They have a dense network of fenestrated capillaries that are seven times more abundant than the peripheral nerves and create a permeable blood–DRG barrier [1,3].

In the DRG, there are two populations of neuronal somata: “large light” neurons and “small dark” neurons [4]. The former transmit proprioceptive and discriminative touch information throughout the posterior columns, while the latter carry thermal and nociceptive signals that are transmitted to the second sensory neurons ascending to the central nervous system throughout the spinothalamic tract [4]. Based on the type of affected neuron, ganglionopathy is divided into two subtypes: large-fiber and small-fiber sensory ganglionopathy [5]. The first diagnostic criteria for sensory ganglionopathy were suggested in 2009 [6]. In 2014, they were validated on a multicenter population by a francophone collaborative study and showed 90% sensitivity and 85% specificity against an expert diagnosis [7]. However, these criteria are based on findings attributed to lesions of large neuronal somata; therefore, ganglionopathy restricted to small neuronal somata cannot be distinguished by these criteria [8]. A definite diagnosis of sensory ganglionopathy can be made only by performing a pathological examination of the DRG; however, this is not recommended in routine clinical practice [7]. Therefore, the clinical diagnosis mostly relies on clinical evidence of the degeneration of sensory neurons, such as non-length-dependent, patchy, asymmetric sensory disturbances commonly localized to the proximal extremities and trunk [2,5]. A considerable number of cases may be left undiagnosed because of their unusual presentation, sometimes suggestive of somatoform disorders [9].

Harlequin syndrome is a rare disorder of the autonomic nervous system that was first described by Lance et al. in 1988 [10]. It is characterized by unilateral flushing and sweating of the face, neck, and upper chest, concurrent with contralateral anhidrosis, an absence of sweating [11]. The incidence is unknown, as only sporadic case reports are presented in the literature [12]. Harlequin syndrome usually originates from a unilateral lesion of T2–T3 sympathetic fibers carrying sudomotor and cutaneous vasodilator innervation to the face [12]. In the case of T1 sympathetic fiber involvement, occulosympathetic innervation will be affected and ipsilateral Horner syndrome may be seen [12]. Regarding its etiology, half of the cases are due to either structural or iatrogenic causes, while the other half are idiopathic [11]. Only a few authors have raised the possibility of an autoimmune mechanism for Harlequin syndrome [13].

One of the systemic diseases that can involve the nervous system is sarcoidosis, a multisystem granulomatous disorder of unknown origin that affects an estimated 2 to 160 people per 100,000 worldwide [14]. In 90% of cases, sarcoidosis affects the lungs and mediastinal lymph nodes, although it can involve virtually any organ [15]. Nervous system involvement, or neurosarcoidosis, clinically manifests in approximately 5% of cases, although as many as one in five patients with sarcoidosis can have asymptomatic lesions [15]. Facial and optic neuropathies are seen most frequently, although any part of the nervous system can be affected [16]. Research shows that up to 30% of patients with sarcoidosis suffer from small-fiber neuropathy [15], whereas there is only anecdotal data about the incidence of ganglionopathy and Harlequin syndrome in this population [17].

In this case report, we present a patient with both sensory ganglionopathy and Harlequin syndrome which is probably attributed to the sarcoidosis. It connects not only the several neurological disorders but also a range of different medical specialties which were essential in the diagnosis and treatment of the patient.

## 2. Case Presentation

A 31-year-old woman presented to the outpatient Neurology Department complaining of burning pain in the right side of the upper chest and feet. She scored the severity of pain in her chest as 6 on the numeric rating scale (NRS). Moreover, she experienced episodes of intense flushing and sweating on the right side of her face, neck, and upper chest (Figure 1) when engaging in strenuous physical activity or spending time outdoors on a warm day. The patient had experienced these symptoms for one year. Gabapentin 300 mg at night was prescribed by a general practitioner; however, the effect was insufficient.

The patient’s first symptoms began 3 years ago, when she started complaining of general weakness, fatigue, and shortness of breath during moderate physical activity. A chest X-ray and computed tomography (CT) scan were performed and revealed hilar and mediastinal lymphadenopathy (Figure 2a). Considering the patient‘s symptoms and radiological findings, she was referred to a pulmonologist. Pulmonary function testing, including spirometry and lung diffusion test, was performed and no abnormalities were detected. Magnetic resonance imaging (MRI) of the neck, chest, abdomen, and pelvis was performed to specify the CT scan findings and differentiate granulomatous disease from lymphoproliferative disorders. MRI confirmed the presence of mediastinal lymphadenopathy and demonstrated small perilymphatic nodules in the perihilar region of the lungs, both of which accumulated contrast agent (Figure 2b). Moreover, hepatosplenomegaly was observed and suspected to be caused by sarcoidosis. To confirm the diagnosis by pathology, a perihilar lymph node biopsy was attempted. However, the biopsy was performed inaccurately and the results were not informative. Therefore, based on the MRI scan results, the final clinical diagnosis of mediastinal and pulmonary sarcoidosis was made. After a year, MRI of the chest was repeated and did not show any new lesions in the lungs or lymph nodes. Considering the stable course and mild symptoms of the disease, no treatment was prescribed and the patient was kept under observation.

In parallel, the patient was thoroughly examined by a cardiologist to rule out a cardiovascular origin of her breathlessness. The echocardiogram showed reduced longitudinal strain of the heart. A Holter monitor revealed second-degree atrioventricular block and short episodes of sinus node arrest. Sarcoidosis-induced cardiomyopathy was suspected and an MRI of the heart was scheduled. However, it was negative for nodular myocardial lesions, focal myocardial thickening, scars, or other abnormalities despite slight hypertrophy of the left ventricle wall. Cardiac sarcoidosis was ruled out and the patient was kept under observation. After 2 years, the echocardiogram and Holter monitor were repeated, and the results showed no progression of cardiovascular pathology. However, the patient complained of flushing and sweating on the right side of the upper body during physical activity. The cardiologist suspected an autonomic nervous system disorder and referred the patient to a neurologist.

The patient’s medical history was also taken into account. The patient suffered from migraine without aura since adolescence. In the past year, she experienced up to six migraine attacks per month. However, the headache was successfully controlled only by symptomatic treatment with sumatriptan and she was not referred to the neurologist for a preventive therapy for migraine. The patient also had primary hypertension and dyslipidemia and was prescribed perindopril 5 mg twice a day and atorvastatin 20 mg daily. Moreover, the patient was a previous smoker with a history of 8 pack years.

When the patient first visited the neurologist, a thorough neurological examination was performed. Both pupils were about the same size and responded to light equally. Cranial nerves were intact. Muscle strength and tone were normal in all extremities. On sensory examination, two patterns of sensory disturbances were identified. In one pattern, the patient noted hypoalgesia in the tips of the fingers and hyperalgesia in the feet consistent with polyneuropathy. In the other pattern, she identified pronounced hyperalgesia and allodynia in the upper part of the right chest and some patches on the right side of the upper back. Vibration sense was intact. The remainder of the neurological examination was normal.

The sensory disturbance patterns led to a preliminary diagnosis of sensory ganglionopathy and small-fiber neuropathy. Even though a causative relation between the neurological disturbances and sarcoidosis was suspected, an extensive diagnostic workup was performed to eliminate other possible causes of these disorders. Complete blood count, erythrocyte sedimentation rate, and levels of C reactive protein, creatinine, urea, creatine kinase, liver enzymes (aspartate transaminase, alanine transaminase, and alkaline phosphatase), thyroid hormones (thyroid stimulating hormone, thyroxine (T4), and triiodothyronine (T3)), and vitamin B12 were within the normal ranges. The glucose tolerance test was unremarkable. Immunological tests, including antinuclear antibody, anti-double-stranded DNA antibody, rheumatoid factor, and paraneoplastic antibody panels, were negative as well.

To determine the origin of complaints, instrumental tests were also carried out. Nerve conduction studies and electromyography, including the cutaneous silent period, showed no abnormalities. MRI of the thoracic spine was performed to rule out a myelopathic or vertebrogenic origin of the sensory disturbances in the chest region. It revealed left paracentric Th7/8 disc and central Th8/9 disc extrusions without spinal cord or nerve root compression.

To identify possible extrapulmonary manifestations of sarcoidosis, the patient was referred to a rheumatologist; however, no signs of other organ involvement were observed (Figure 3).

As the comprehensive laboratory and instrumental evaluation of the patient revealed no abnormalities, the final diagnosis was based on the patient’s complaints and the findings on neurological examination. Asymmetrical, non-length-dependent patterns of hyperalgesia and allodynia in the chest were attributed to sensory ganglionopathy. The burning sensation in the feet and sensory disturbances in the fingers and feet confirmed the diagnosis of small-fiber neuropathy. The unilateral flushing of the face, chest, and arm was attributed to Harlequin syndrome due to the contralateral lesion of sympathetic fibers. Since many other possible metabolic, immunological, or structural causes were ruled out, the nervous system lesions were presumed to originate from a sarcoidosis-induced dysimmune mechanism. Thus, the final clinical diagnosis of sarcoidosis-induced sensory ganglionopathy, small-fiber neuropathy, and Harlequin syndrome was made.

Treatment for neuropathic pain control was initiated during the first neurology appointment. First, pregabalin was prescribed, starting with 75 mg at night. After a week, during which the patient was consulted repeatedly, she complained of an insufficient treatment effect, leading to an increase in the pregabalin dose to 150 mg twice a day. Moreover, amitriptyline was initiated as it was also suitable for the prophylactic treatment of migraine. In addition, a 7-day course of oral dexamethasone (1 mg twice a day for 3 days, followed by 0.5 mg twice daily for 4 days) was administered. After 2 weeks, the patient’s symptoms remained unchanged, and the pregabalin dose was increased to 150 mg twice a day and the amitriptyline to 25 mg twice a day (Figure 4). After 2 months, the patient reported remission of the burning sensation in the feet, meaning that the symptoms of small-fiber neuropathy were successfully controlled. However, the pain in her upper-right chest region remained almost unchanged, suggesting that the ganglionopathy-induced symptoms were resistant to the prescribed treatment. For further treatment, the patient was hospitalized in the Neurology Department.

When the patient was admitted as an inpatient, therapeutic plasma exchange as an immune-modulating treatment was selected. A 5-day course of daily procedures was completed. After this treatment, the patient reported a slight reduction in the pain in her chest (NRS score 4). On discharge, amitriptyline 25 mg twice a day and pregabalin 150 mg twice a day were prescribed for long-term pain modulation (Figure 4). Considering the need of immunosuppressive treatment, the patient was monitored by a pulmonologist and rheumatologist and it was not recommended as no other evidence of systemic disease activity was detected.

## 3. Discussion

This report describes a rare case of sarcoidosis-associated sensory ganglionopathy, small-fiber neuropathy, and Harlequin syndrome. It illustrates a unique set of sensory and autonomic symptoms, probably induced by a dysimmune mechanism.

Sensory ganglionopathy is characterized by sensory deficits in the trunk or proximal limbs that present in asymmetrical and patchy patterns [1]. In the presented case, the patient’s complaints and neurological exam findings revealed right-sided hyperesthesia and allodynia in the upper thoracic region, suggesting small-neuron lesions in the DRG at this level. In such cases, when small DRG cells are selectively involved, confirming the diagnosis is complicated and remains mainly clinical [8], as the diagnostic criteria for sensory ganglionopathy are based on symptoms and test findings attributed to large-fiber damage [6]. Nerve conduction studies are not informative, as nerve fibers smaller than 7–10 μm do not contribute to sensory nerve action potentials due to their slow rate of conduction [18]. However, it is essential to rule out the possibility of structural abnormalities as vertebrogenic pathology may sometimes cause a similar pain pattern. Thus, in our case, both NCS and thoracic spine MRI were performed and showed no abnormalities.

An alternative diagnosis explaining the patient’s symptoms might be non-length-dependent small-fiber neuropathy (NLD-SFN). Currently, it is under dispute whether small-neuron ganglionopathy and NLD-SFN are the same disorder or two distinct diagnostic entities [19]. Like small-fiber ganglionopathy, NLD-SFN is not defined in the diagnostic criteria, which are adjusted to the length-dependent pattern of small-fiber damage [20]. However, NLD-SFN is frequently associated with systemic disease, and sarcoidosis is one of the most common reasons for dysimmune NLD-SFN [19], as in our case. In fact, sarcoidosis-induced NLD-SFN is separately classified as paraneurosarcoidosis because it is thought to be induced by circulating inflammatory mediators as opposed to structural damage by granulomas [15]. The latter mechanism can also induce small-fiber lesions because the formation of perineurial granulomas involves small cutaneous nerves [21]. According to pathological studies, perineurial granulomas are identified in approximately half of patients with sarcoidosis [21]. However, in our case, the physical examination of the patient did not reveal any skin lesions or subcutaneous formations in the affected areas. Moreover, the neurological exam revealed two distinct patterns of sensory disturbances: length-dependent symptoms in the fingers and feet and non-length-dependent hyperalgesia in the right side of the chest and scapular area. However, only a histological examination can give a definite answer, but DRG biopsy is not recommended [6]. Therefore, more pathological and histological studies are needed to exclude small granulomatous lesions of the small nerve fibers and dorsal root ganglia and improve the classification and diagnostics of sensory ganglionopathy and NLD-SFN.

In the search for an etiological agent, the clinician should perform a detailed investigation of the most common and even life-threatening causes, such as paraneoplastic syndrome. As nearly a third of ganglionopathy cases are related to an underlying oncological disorder [2], we performed a paraneoplastic antibody panel on our patient’s blood, and it came back negative. Many other blood tests were performed, which ruled out the possibility of hepatic or thyroid dysfunction, diabetes mellitus, vitamin B12 deficiency, underlying inflammation, or systemic rheumatological disorders. The absence of other abnormalities and the patient’s history of pulmonary sarcoidosis led to the conclusion that a dysimmune mechanism attributed to sarcoidosis was the most likely etiological factor in terms of nervous system involvement.

The exact mechanisms Involved in the sarcoidosis provoked immunological reactions is still under investigations. In the described clinical cases, dysimmune sensory ganglionopathy most commonly occurs with Sjögren’s syndrome, celiac disease, systemic lupus erythematosus, primary biliary cholangitis, and autoimmune hepatitis [2,5]. To date, sarcoidosis-induced sensory ganglionopathy was suspected only in a few patients described in the larger case series and compose a total of five cases [22,23,24,25]. However, a causal link between these autoimmune disorders and sensory ganglionopathy remains to be proven [1]. Currently, a new antibody reacting with the intracellular domain of the fibroblast growth factor receptor 3 (FGFR3) was identified in a subgroup of patients with sensory ganglionopathy, including two patients with sarcoidosis [23]. Moreover, antibodies against the family of argonaute proteins were discovered in the group of 21 neurological patients, among which, the main clinical presentation was sensory neuronopathy (38%) and one of them had comorbidity with sarcoidosis [24]. These findings suggest that various underlying autoimmune diseases, including sarcoidosis, may provoke the production of antibodies targeting the dorsal root ganglia. Further research of serologic biomarkers of sensory ganglionopathy is essential to shed light on the immunological mechanisms of this disorder, improve its diagnostics, and find new targets for possible treatment methods.

Only a few case reports raised the possibility of an autoimmune origin of Harlequin syndrome. Karam [13] published the case of a 76-year-old man who developed both Harlequin and Horner syndromes as a part of autoimmune autonomic ganglionopathy. A few other cases were attributed to possible autoimmune mechanisms due to autoimmune hyperthyroidism [26] and Guillain–Barre syndrome [27]. To the best of our knowledge, this is the second described case of sarcoidosis-induced Harlequin syndrome. Wong et al. [17] presented the case of a 61-year-old woman with systemic sarcoidosis that manifested as Harlequin and Horner syndromes. In that case, the clinical presentation of sympathetic dysfunction overshadowed the most common respiratory and nonspecific symptoms of sarcoidosis such as cough, dyspnea, fatigue, fever, and weight loss. Conversely, in our case, nervous system involvement manifested 2 years after the diagnosis of pulmonary sarcoidosis. The etiology of sarcoidosis-associated Harlequin syndrome may be explained by several mechanisms. In Wong et al.’s case, they supposed it was caused by compression of the sympathetic chain due to parenchymal lung disease or mediastinal lymphadenopathy [17]. In our case, the patient’s radiological studies also showed enlarged mediastinal lymph nodes. However, Harlequin syndrome developed later in the course of the disease and manifested at a time nearly concurrent with ganglionopathy and small-fiber neuropathy, suggesting that it may have been triggered by the same etiological agent. Therefore, we cannot deny either a structural or dysimmune mechanism for Harlequin syndrome.

Moreover, a relationship between Harlequin syndrome and headache was suggested in the literature. To date, approximately 90 cases of Harlequin syndrome are reported in the literature [28] and 11 of them showed comorbidity with headaches [28,29]. Migraine is the most common headache in this population; however, pathological association between these two disorders is doubtful. In most presented cases as well as in our case, migraine started years before the development of Harlequin syndrome and autonomic symptoms were not simultaneous with headache attacks [28,30]. Two case reports raised the possibility of the association between cluster and Harlequin syndrome [28,31]. Although, a dysfunction of the sympathetic nervous system is a plausible mechanism for both disorders, a shared underlying structural cause cannot be fully excluded in these cases [28]. Therefore, currently, there is a lack of data on a direct link between headaches and Harlequin syndrome.

Treating sensory ganglionopathy is challenging, as no randomized controlled trials are investigating it and the choice is commonly based only on successful descriptive case series. In general, neurosarcoidosis treatment depends on the severity of case. In mild cases, such as isolated facial nerve palsy, oral corticosteroids may be sufficient [15,32]. In moderate and severe cases, including meningitis, multiple cranial neuropathies, or optic neuropathy or myelopathy, high dose corticosteroids intravenously followed by immunosuppressive medications, such as methotrexate or mycophenolate mofetil, should be administered [15,16]. In the most severe and treatment resistant cases, infliximabe, anti-tumor necrosis factor alpha monoclonal antibody, or cyclophosphamide should be considered [15,32]. Observational data suggest that sarcoidosis-induced ganglionopathy and small-fiber neuropathy are resistant to standard care [33]. Immunomodulatory therapy is essential, and the best results can be achieved by administering intravenous immunoglobulin (IvIg) alone or combined with immunosuppressive medications [33,34]. Some authors suggest that immunosuppressive treatment may be administered to patients specifically for neuropathic pain in the absence of active systemic disease to prevent the development of chronic pain syndrome [33]. However, a benefit–risk ratio should be carefully evaluated as immunosuppressive medications carry a risk of significant side effects such as opportunistic infections, bone marrow suppression, hepatotoxicity, and others. Moreover, the best results have been observed when patients were treated within the first 2 months after the onset of symptoms, while only stabilization of the disease was achieved in those treated within the first 8 months [35]. In our case, the patient first came to the neurologist already a year into her symptoms. Her treatment was started with neuropathic pain management and glucocorticoids. However, it showed no effect on ganglionopathy-related pain. Therefore, therapeutic plasma exchange was selected as a second-line treatment option (IvIg administration for this indication is restricted by the national health policy in our country) and it led to partial pain relief. These findings suggest that plasma exchange can be an option in mild or moderate cases or when more effective treatments, such as IvIg administration, are unavailable due to high costs or other reasons.

Harlequin syndrome is usually a benign condition and its treatment depends on the severity of the symptoms [12]. In the case of troublesome flushing causing social embarrassment, a stellate ganglion blockade or botulinum toxin injection is recommended [12,36]. Surgical sympathectomy may be considered in severe cases [12]. In the case of an autoimmune mechanism, Harlequin syndrome can also be successfully controlled with high-dose glucocorticoids, administered either orally or intravenously [13,17]. In the presented case, the unilateral flushing and sweating were episodic, and no specific treatment was administered to the sympathetic nervous system.

Our case report has some limitations. First, we only presented one clinical case. Moreover, there is a lack of experience in the diagnosis and treatment of these disorders, which led to the incomplete recovery of the patient.

## 4. Conclusions

This clinical case illustrates a wide variety of neurological disorders in the presence of sarcoidosis, including sensory ganglionopathy, small-fiber neuropathy, and Harlequin syndrome. The exact mechanism of nervous system involvement in cases of sarcoidosis remains to be elucidated. As a dysimmune mechanism is the most likely possibility, structural damage due to perineurial granulomas and enlarged lymph nodes cannot be ruled out. This implies that an autoimmune etiology should be suspected in the case of DRG and autonomic nervous system involvement. Therefore, based on this case, we strongly suggest that the diagnostic workup of these patients should include screening for the most common autoimmune disorders, such as sarcoidosis, Sjögren’s syndrome, systemic lupus erythematosus, and others. An accurate and timely diagnosis is essential for the selection of treatment and early initiation of immunomodulatory therapy in cases of an underlying dysimmune mechanism in order to achieve adequate control of the disease.

## Figures and Tables

**Figure 1 medicina-59-01495-f001:**
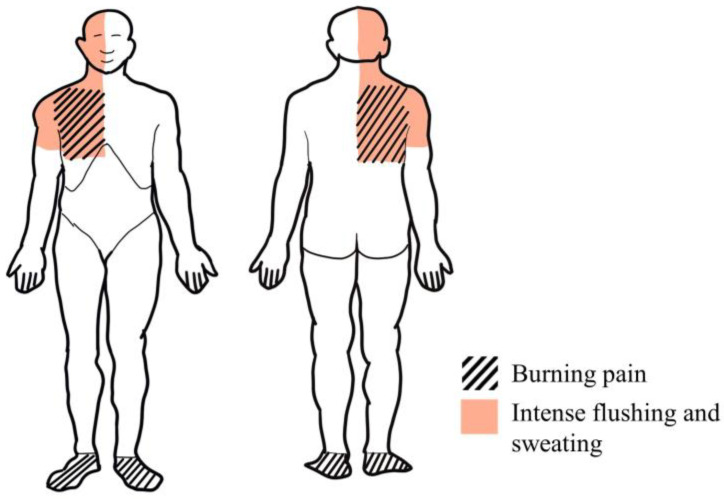
Schematic illustration of patient’s complaints and findings of neurological examination.

**Figure 2 medicina-59-01495-f002:**
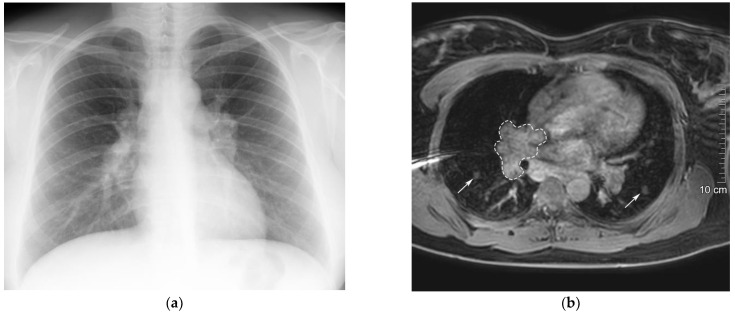
Radiological examinations of the patient’s chest. (**a**) Chest X-ray showing mediastinal lymphadenopathy. (**b**) Chest MRI showing gadolinium-enhanced pulmonary nodules (arrows) and mediastinal lymphadenopathy (dashed line) in mDIXON-W MRI sequence 5 min after gadolinium injection.

**Figure 3 medicina-59-01495-f003:**
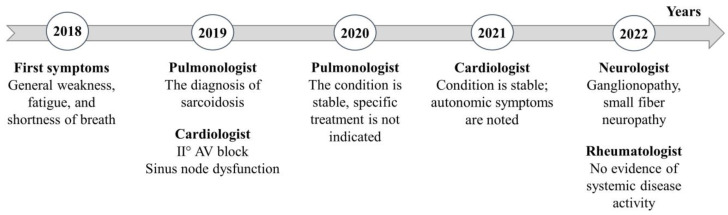
Schematic presentation of disease course and examination of patient by different medical specialists. Abbreviations: AV—atrioventricular.

**Figure 4 medicina-59-01495-f004:**
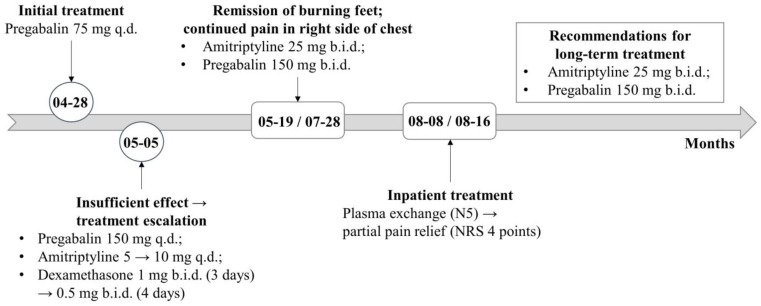
Schematic presentation of patient‘s treatment. Abbreviations: q.d.—once a day; b.i.d.—twice a day; NRS—numeric rating scale.

## Data Availability

The data presented in this study are available from the corresponding author.

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
