# Peer review of "Sarcoidosis-Associated Sensory Ganglionopathy and Harlequin Syndrome: A Case Report"

_medicina, 2023, doi:10.3390/medicina59081495_

Round 1
Reviewer 1 Report
The case report illustrates a wide variety of neurological disorders in the presence of sarcoidosis. It helps to figure out dysimmune mechanism to improve therapy. However, several points need to be improved:
1> Please use high-resolution figures in article. The words in Figures 3 and 4 are unclear. Adding notes to some of the abbreviations used in the figure will help the reader to understand better.
2> Please give a more detailed explanation of why certain treatments did not work.
Some side effects of treatment should be mentioned.
3> Please discuss further at the end of the report how we could improve treatment based on this case.
Reviewer 2 Report
Thank you for inviting me to review the case report “Sarcoidosis-associated sensory ganglionopathy and Harlequin syndrome: a case report”. The manuscript is generally well-written, but there are sections where the organization could be improved for better flow.
1. The introduction provides a good background on sensory ganglionopathies and their association with autoimmune mechanisms, but it could benefit from a clearer transition to the specific case being presented.
2. The case presentation is comprehensive and includes relevant details about the patient's symptoms, medical history, diagnostic workup, and treatment. However, it might be beneficial to include more specific details about the patient's history of migraine and its relevance to the overall presentation.
3. The discussion section is thorough and provides a detailed analysis of the case. However, it could be more concise and focus on the key points, especially regarding the diagnostic challenges and treatment options. Consider highlighting the novel aspects of the case and how it contributes to the existing literature.
4. While the manuscript discusses possible mechanisms for the observed conditions, it might be useful to provide a more in-depth discussion of the potential immunological processes involved in the dysimmune mechanism, given the central role this mechanism plays in the case.
5. The manuscript could benefit from a more comprehensive comparison with existing literature on sarcoidosis-associated neurological manifestations, including sensory ganglionopathies and Harlequin syndrome. Discuss how this case adds to the current understanding of these conditions and their relationship to sarcoidosis.
6. Address the limitations of the study more explicitly. Discuss potential future research directions that could further explore the underlying mechanisms, diagnostic approaches, and treatment strategies for similar cases.
7. Some sentences are quite long and could be broken down for improved clarity. Additionally, certain technical terms could be defined or explained further for readers who may not be experts in the field.
8. Mention any ethical considerations, such as patient consent for publication of their medical information, if applicable.
9. Summarize the key takeaways from the case and its implications for clinical practice and research. Emphasize the importance of early diagnosis and appropriate management in similar cases.
Overall, the manuscript provides a valuable clinical case that contributes to the understanding of neurological manifestations of sarcoidosis. With some organization and clarity improvements, along with a more focused discussion, it has the potential to be a significant addition to the literature in this field.
Moderate English editing is required.
Reviewer 3 Report
The subject is very uncomon,I really read it with intetest.
I ask the auhors if they perform and cervical and thoracal MRI with contrast?
Or a brain MRI with contrast?
PET CT scan was performed . It could be use, maybe to predicte the re-flare , however, the reoccurrence is common with the stimulus. The underlying disease process is the main determinant in predicting the patient's long-term outlook with Harlequin syndrome, i Think the patient needs immunosupresion-, I recommend Azathioprine as rheumatologist usually in neurological manifestation . but as well, metothrexate or Mycophenolate mophetil could be an option. But mandatory imunosupresion in my clinical opionion
More data in discussion about the treatment option for lonf time in neurosarcoidosis
Good luck
-
Round 2
Reviewer 2 Report
Thank you for responding to all of my comments. I have no further comments.
I have no more comments.
Reviewer 3 Report
introduction- a graphical representation of neurological manifestation ( mechanism)